

# Drivers of microplastic pollution in soil sediments at fish landing centers in Sri Vijaya Puram (Port Blair), South Andaman Island

Ajit Kumar[1,*], Akshatha Soratur[1,*], Sumit Kumar[2], R Kiruba-Sankar[3], Dilip Kumar Jha[4] and Balu Alagar Venmathi Maran[5]

[1] Department of Ocean Studies and Marine Biology, Pondicherry University, Sri Vijaya Puram, Andaman and Nicobars Islands, India

[2] Department of Industrial Fish and Fisheries, Babasaheb Bhimrao Ambedkar Bihar University, Muzaffarpur, Bihar, India

[3] ICAR-Central Island Agricultural Research Institute, Sri Vijaya Puram, Andaman and Nicobar Islands, India

[4] Atal Centre for Ocean Science and Technology for Islands, National Institute of Ocean Technology (NIOT), Sri Vijaya Puram, Andaman and Nicobar Islands, India

[5] Graduate School of Integrated Science and Technology, Nagasaki University, Bunkyomachi, Nagasaki, Japan

[*] These authors contributed equally to this work.

Corresponding author
Balu Alagar Venmathi Maran, bavmaran@nagasaki-u.ac.jp

## ABSTRACT

Microplastic pollution poses a growing global threat to marine ecosystems, and soil sediments at fish landing centres are an often-overlooked reservoir of microplastics. Sri Vijaya Puram (Port Blair), located in South Andaman Island, is critical for fisheries and marine biodiversity, making it an important area for studying microplastic pollution. This study aims to identify the key drivers of microplastic pollution in soil sediments at fish landing centers. The specific objectives included assessing microplastic abundance, characterizing polymer types, and identifying potential pollution sources, such as fishing gear, plastic packaging, and urban runoff. Sediment samples were collected from six fish landing centres such as Junglighat, Dignabad, Chatham, Guptapara, Wandoor and Chidiyatapu. The study revealed significant spatial variation in microplastic concentrations, with higher contamination in the northern region. Notably, the northern region (centers) had a significantly higher mean abundance of $251.4 \pm 110.3$ particles/kg compared to the southern region's (centers) $105.0 \pm 57.1$ particles/kg. The Mann–Whitney $U$ Test ($U = 283.0$, $p$-value $= 0.00014$) substantiated this significant difference. Dominant polymer types included aramid fiber, acrylonitrile-butadiene rubber, and polyisoprene, indicating industrial and consumer waste sources. Potential sources were linked to urban runoff, fishing activities, and inadequate waste management. This study contributes to understanding microplastic pollution drivers in tropical coastal environments. The findings highlight the impact of anthropogenic activities and land use patterns on microplastic pollution in fish landing centres. This information is crucial for developing targeted mitigation strategies in similar coastal regions.

---

## INTRODUCTION

Microplastic pollution has emerged as a critical global environmental issue, with an estimated 381 million tons of plastic produced annually, a significant portion of which degrades into microplastics (<five mm) through fragmentation, industrial abrasion, and the release of synthetic fibers (*Anuar et al., 2022*; *Khan & Bose, 2024*; *Andrady, 2011*). The ubiquity of microplastics in marine, freshwater, and terrestrial ecosystems has raised significant concerns among scientists and policymakers worldwide (*Bergmann et al., 2019*). Studies have extensively documented the presence of microplastics in oceans, rivers, and soil sediments, highlighting their persistence and accumulation over time. Microplastics have been found in marine environments, including the deepest parts of the ocean, such as the Mariana Trench, where they accumulate in sediments and water columns (*Peng et al., 2018*). Similarly, freshwater ecosystems, including rivers and lakes, show significant contamination, with microplastics observed in both surface waters and sediments (*Dris et al., 2015*; *Osorio, Tanchuling & Diola, 2021*). Soil sediments, particularly in floodplains and agricultural areas, act as reservoirs for microplastics, with deposition influenced by environmental factors like flood dynamics and bioturbation (*Weber et al., 2021*). Recent estimates suggest that millions of metric tons of plastic debris enter the environment annually, a substantial portion of which eventually breaks down into microplastics (*Jambeck et al., 2015*). These particles pose risks not only to aquatic and terrestrial organisms but also to human populations through food chain contamination, particularly in regions dependent on fisheries and coastal resources (*Rochman et al., 2015*). Coastal environments are particularly susceptible to microplastic pollution due to the convergence of land-based and marine sources. In fishery-dominated regions, microplastic contamination is exacerbated by anthropogenic activities such as fish processing, waste disposal, and maritime transport (*Lusher, Hollman & Mendoza-Hill, 2017*). The concentration of microplastics in soil sediments at fish landing sites is an emerging area of concern, as these locations serve as critical interfaces between terrestrial and aquatic ecosystems (*Arab, Yu & Nayebi, 2024*). Microplastics can enter soil sediments through various pathways, including atmospheric deposition, surface runoff, and direct human activities such as improper waste disposal and fishing gear degradation (*Büks & Kaupenjohann, 2020*). Soil sediments act as long-term reservoirs for microplastics, influencing their distribution, degradation, and potential ecological risks (*Corradini et al., 2019*). Understanding the extent of microplastic accumulation in such environments is essential for assessing its ecological consequences and informing management strategies.

Fish landing centers, characterized by high human activity and direct interaction with marine resources, represent potential hotspots for microplastic contamination. These sites are critical nodes in the fisheries supply chain, where fish are landed, sorted, processed, and transported. The accumulation of microplastics in soil sediments at these locations may stem from multiple sources, including discarded fishing nets, plastic packaging, and residual waste from fish handling and cleaning operations (*Barboza et al., 2018*). Additionally, microplastics from fishing vessels, lost gear, and shoreline litter may be transported to fish landing areas *via* tidal currents and storm surges, further exacerbating

pollution levels (*Sterl, Delandmeter & Van Sebille, 2020*). The high influx of plastic waste in these areas can contribute to the degradation of coastal ecosystems and pose risks to marine biodiversity. Sri Vijaya Puram (Port Blair), the administrative capital of the Andaman and Nicobar Islands, India, represents a critical locus for investigating microplastic pollution in coastal ecosystems, given its dual role as a biodiversity hotspot and a hub of artisanal and commercial fisheries (*Goswami, Vinithkumar & Dharani, 2020*). The region hosts key fish landing centers, which sustains over 7,000 licensed fishers and a fleet of 2,888 mechanized and non-mechanized vessels, collectively contributing ~43,000 metric tons annually to India's marine fish production (*Kaliyamoorthy, Roy & Sahu, 2022*; *Marine Products Export Development Authority (MPEDA), 2023*). Escalating anthropogenic pressures, such as unregulated tourism, maritime traffic and inadequate solid waste management infrastructure have amplified the introduction and persistence of plastic debris in coastal environments (*Hossain et al., 2022*). Microplastic pollution poses serious ecological and public health risks by disrupting soil properties, impairing nutrient cycling, and infiltrating marine food webs through ingestion by commercially important fish, leading to toxin bioaccumulation and trophic transfer of pollutants (*Rochman et al., 2015*; *Soratur et al., 2024*; *Zhang et al., 2024*). These particles cause physiological harm in marine organisms and threaten fishery sustainability (*De Sá et al., 2018*; *Horn, Granek & Steele, 2019*). They also adhere to algal surfaces, altering their ecological roles and biomedical potential (*Kumar et al., 2025*). Acting as vectors for pathogens and degrading agricultural soils, microplastics demand urgent assessment at fish landing centers to guide mitigation and protect food security (*Barboza et al., 2018*). Sediments in fish landing zones are particularly susceptible to microplastic accumulation, deriving from degraded fishing gear (nylon nets, polystyrene floats), single-use packaging, antifouling paint particles from vessels, and terrestrial runoff during monsoon seasons (*Xue et al., 2020*; *Govender et al., 2020*). These microplastics pose significant ecological threats, including bioaccumulation in commercially vital species such as tuna (*Thunnus* spp.) and groupers (*Epinephelus* spp.), which constitute a primary protein source for local communities (*Malakar, Venu & Kumar, 2019*). Furthermore, mangrove ecosystems adjacent to fishing hubs, critical for carbon sequestration, shoreline stabilization, and juvenile fish nurseries are also at risk of sediment contamination, potentially disrupting benthic communities and microbial processes (*Kiruba-Sankar et al., 2025*).

In this study, we investigated the prevalence and characteristics of microplastics in soil sediments at fish landing centers in Sri Vijaya Puram (Port Blair), South Andaman Island, an area experiencing increasing anthropogenic pressure due to urbanization, tourism, and maritime activity. While previous studies have quantified microplastic presence in coastal sediments, few have linked their spatial distribution to localized drivers such as land use patterns, population density, and waste management efficiency, especially in small island contexts. This study addresses this gap by examining how microplastic abundance and composition vary between the northern and southern regions of South Andaman and identifying the key environmental and human-related factors responsible. Through analyses of microplastic concentrations, polymer types, and potential sources, our research provides a spatially contextualized understanding of contamination in these high-activity

zones. The findings contribute to developing more targeted, region-specific strategies for plastic waste management and coastal conservation in tropical island settings.

## MATERIALS & METHODS

### Site description and sampling

The Andaman and Nicobar Islands (ANI) situated in Bay of Bengal known for their high biodiversity and distinct ecosystems. South Andaman, Sri Vijaya Puram (Port Blair) which is the administrative headquarter of this archipelago is the most populous region. In this region, tropical climate prevails with an annual rainfall of 3,000–3,800 mm from the southwest and northeast monsoons. Rapid population growth, urbanization and tourism have brought great environmental problems including marine plastic pollution (*Krishnakumar et al., 2020*). Sampling was carried out at six distinct fish landing centers in Sri Vijaya Puram, comprising three sites in the northern region (Junglighat, Dignabad and Chatham) and three in the southern region (Guptapara, Wandoor and Chidiyatapu). At each location, six sediment samples (200 g each) were collected three from the high tide line (HTL) and three from the low tide line (LTL). Sampling points were spaced 150 m apart along the beach. A quadrat measuring 25 cm × 25 cm was placed at each sampling location, and the uppermost one cm of sediment was carefully scooped using a metal spoon, following the method outlined by *Dowarah & Devipriya (2019)*. The collected sediment samples were stored in paper bags and transported to the laboratory for further analysis.

### Microplastic analysis

For quantitative analysis, sediment samples (200 g each) were oven-dried at 40 °C for 24 h, following the protocol outlined in *Masura et al. (2015)*. Microplastic particles larger than one mm were manually separated from the dried samples. The subsamples were treated for organic material digestion by adding 20 mL of 30% hydrogen peroxide and 0.05 M iron (II) solution. The mixture was heated to 75 °C for 30 min using a magnetic stirrer. After heating, the solution was transferred into a glass beaker and covered with aluminium foil to prevent contamination (*Peng et al., 2018*; *Masura et al., 2015*). The remaining sediment underwent density separation using a sodium chloride solution (density 1.2 g/cm$^3$) to facilitate the separation of lighter materials from heavier sediment. After allowing the heavier particles to settle, the supernatant was passed through metal sieves with three mm and five mm mesh sizes. The remaining sediment was subjected to a second round of density separation using sodium iodide solution (density 1.8 g/cm$^3$) to isolate microplastics further. The supernatant from this second process was filtered using Whatman membrane filters with a pore size of 0.2 μm, based on the method proposed by *Hidalgo-Ruz et al. (2012)*. The collected microplastic particles were manually examined visually under a stereomicroscope. Four types of microplastics were distinguished, line/fibre (thin fibrous particles and thin straight segments), pellet (hard, sharp fragment), foam (delicate, sponge-like particle) and film (thin flexible plastic planes) (*Free et al., 2014*). Visual characteristics such as colour, brightness, size, and shape were noted to aid in identifying the possible sources of the microplastic particles. For qualitative analysis, Fourier Transform Infrared Spectroscopy

(FTIR) with an attenuated total reflectance (ATR) accessory was employed to identify the polymer composition of the microplastics.

## Quality assurance/quality control

Stringent quality assurance measures were implemented to prevent plastic particle contamination during the experiments. Cotton clothing was worn exclusively by researchers throughout the microplastic extraction process to minimize the risk of contamination (*Lusher et al., 2014*). To further reduce exposure to airborne plastic particles, conical flasks were covered with aluminum foil during the tests, and the laboratory workspace was regularly cleaned with acetone. Following the preventive measures windows and ventilation systems were also kept closed to lessen the possibility of external plastic contamination (*Masura et al., 2015*). Potassium hydroxide (KOH) solutions were prepared, filtered, and inspected under a fluorescent microscope after being stained with Nile red to detect microplastic contamination. Procedural blanks were also analyzed and confirmed to be free of microplastic contamination, ensuring the accuracy of the experimental results.

## Data analysis and polymer identification

Shapiro–Wilk tests were conducted to assess the normality of the data sets. The results indicated a mix of normally and non-normally distributed data. Therefore, both the mean and median were used to represent microplastic abundance in the study areas. The total microplastic abundance across beaches was calculated by combining the HTL and LTL samples from each beach. Non-parametric Mann–Whitney U tests were subsequently performed to evaluate significant differences in microplastic abundance between the North and South regions, as well as between HTL and LTL sampling zones. Further, non-metric multidimensional scaling (NMDS) was performed using a Bray–Curtis dissimilarity matrix to visualize whether the shape composition of microplastics differed between the northern and southern regions. This was followed by Permutational Multivariate Analysis of Variance (PERMANOVA) with 999 permutations, conducted at a significance level of $p < 0.05$, to test whether the two regions differed significantly in terms of their shape composition. The ESA WorldCover 10 m v200 dataset was utilized to assess the land use and land cover (LULC) composition of the study area. A one km buffer was generated around each sampling point to extract and quantify the areas covered by various LULC categories. These data were then integrated with the GHS population grid (R2023) and used to perform a principal component analysis (PCA) to identify the influence of different LULC types and population density on microplastic abundance. All statistical analyses and mapping were done using R and QGIS, respectively. Polymer identification was conducted using Fourier Transform Infrared Spectroscopy (FTIR) with an attenuated total reflectance (ATR) accessory (Intertek Model ASTM E168). This method measures molecular vibrations caused by chemical bonds within the 400–4,000 cm$^{-1}$ infrared spectrum, corresponding to wavelengths of 2.5 to 25 $\mu$m ($10^{-3}$ mm) (*Renner, Schmidt & Schram, 2018*). Individual microplastic particles (ranging in size from 1.5 mm to five mm) were placed onto the ATR diamond crystal using metal needle- nose forceps. A minimum force of 80 N was applied to ensure solid contact between the sample and the ATR crystal, allowing for accurate chemical composition analysis (*Karthik et al., 2018*).

## RESULTS

### Spatial variation in microplastic abundance

The study found that there is significant spatial variation in microplastic presence among fish landing centers in Sri Vijaya Puram (Port Blair), South Andaman Island with a major regional difference between north and south region (Fig. 1). The overall mean abundance of microplastics was $178.2 \pm 103.3$ particles/kg (median = 148.7, IQR = 178.5), whereas regional comparisons revealed substantial variations. The North exhibited significantly higher contamination (Fig. 2) (mean = $251.4 \pm 110.3$ particles/kg, median = 242.5, IQR = 191.25) compared to the South (mean = $105.0 \pm 57.1$ particles/kg, median = 80.0, IQR = 96.25), confirmed by a Mann–Whitney U test ($U = 283.0$, $p = 0.00014$). Intra-regional heterogeneity was observed in the North where Chatham (386 particles/kg) and Janglighat (370 particles/kg) comprised the hotspots, different from Dignabad higher contamination, (149 particles/kg; SD range: 13.97–16.50). Conversely, southern stations displayed reduced loads with Wandoor being the least polluted (66 particles/kg), followed by Guptapara (104 particles/kg) and Chidiyatapu (208 particles/kg). Notably, Chidiyatapu's microplastic levels mirrored mid-range northern values, suggesting localized anthropogenic inputs despite broader regional trends. Replicate-level variability further highlighted differential contamination drivers, with minimal fluctuations at Wandoor (SD = 0.89) compared to Guptapara (SD = 6.87) and Chatham (SD = 13.97).

Habitat-specific analysis (Fig. 3) showed higher microplastic abundances at the high tide line (HTL: mean = $198.6 \pm 123.8$ particles/kg, median = 157.5, IQR = 160.0) than the low tide line (LTL: mean = $157.8 \pm 102.75$ particles/kg, median = 125.0, IQR = 166.25), though this difference was statistically insignificant (Mann–Whitney $U = 196.5$, $p = 0.282$). Northern HTL-LTL contrasts were pronounced, exemplified by Chatham (HTL: 205 particles/kg *vs.* LTL: 181 particles/kg) and Janglighat (HTL: 215 particles/kg *vs.* LTL: 155 particles/kg). In contrast, southern stations demonstrated mixed patterns, Wandoor showed identical HTL and LTL abundances (33 particles/kg), while Guptapara's HTL (69 particles/kg) nearly doubled its LTL (35 particles/kg), implicating tidal deposition and sediment retention mechanisms. The absence of overall HTL-LTL significance likely reflects regional counterbalances, where stark northern contrasts offset southern uniformity. North had more microplastics (2.4-fold $p < 0.001$) than South, highlighting an increased exposure due to proximity with urban centers, maritime exerture and hydrological modulation. Provenance of contamination from the above suggests that localized variations indicate geophysical mechanisms involved and/or anthropogenic impacts that are habitat-specific retention capacities driven.

### Shape and colour

Microplastic composition across six fish landing centers in Sri Vijaya Puram (Port Blair) exhibited pronounced variability in particle types and color distributions (Fig. 4). Five microplastic shapes, films, fragments, fibers, foam and paint particles were identified (Table 1), with fragments dominating the total load (358 particles, 27.90%). Fragment abundance varied spatially, peaking at Chidiyatapu (103 particles, 49.5%) and minimizing at Wandoor (11 particles, 16.7%). Fibers ranked second (279 particles, 21.75%), showing

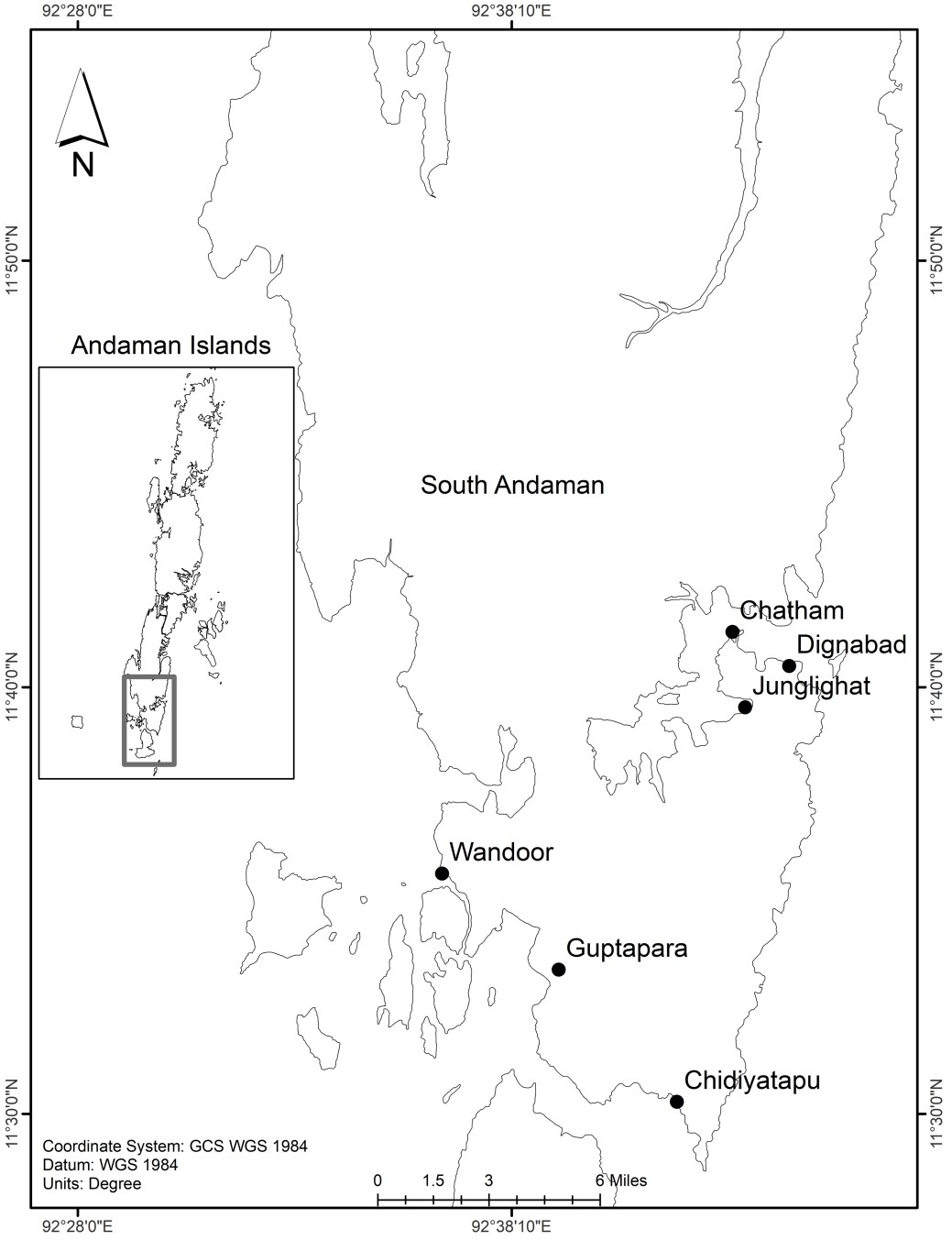

**Figure 1 Map of South Andaman Island, India, indicating the locations of sediment sampling sites.** The sampling locations in South Andaman Island, which include Chatham, Dignabad, Junglighat, Wandoor, Guptapara, and Chidiyatapu. These marked points represent key coastal and inland sites where environmental or biological data were collected for research purposes.

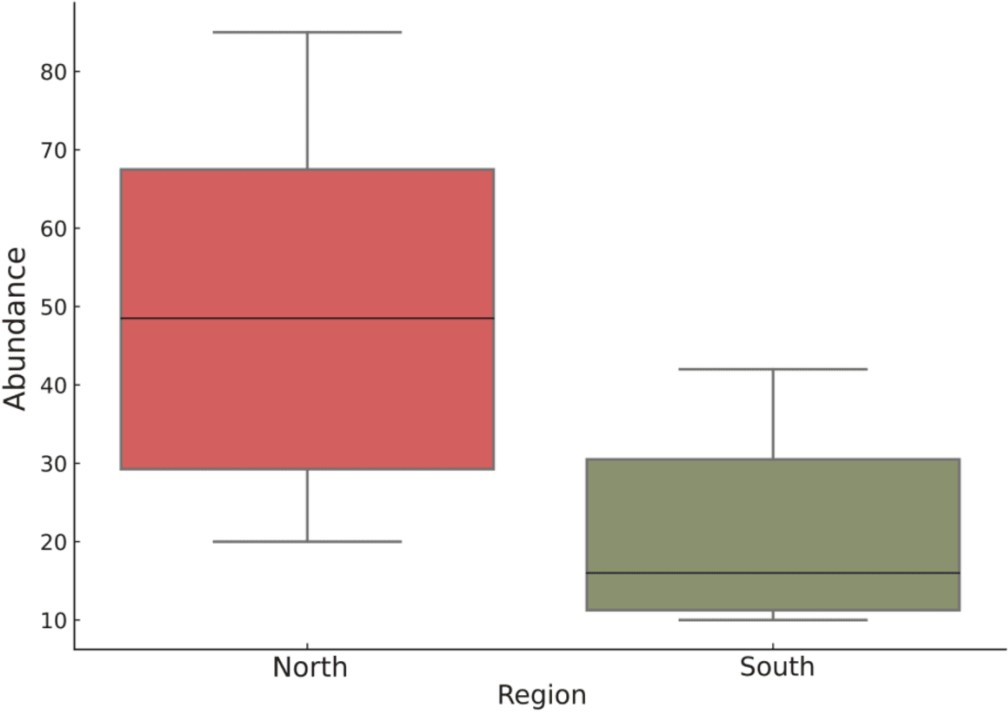

**Figure 2 Boxplot comparing microplastic abundance (particles per 200 g) between the North and South regions of the study area.** The distribution of abundance values for each region, highlighting the median, interquartile range (IQR), and potential variability within the data. In the North region, species abundance shows a higher median and broader range, suggesting greater ecological richness and variability. Conversely, the South region exhibits a lower median and more compact distribution, indicating lower overall abundance and less variation among samples. These differences may reflect regional disparities in habitat quality, resource availability, or environmental stressors affecting species populations.

elevated contributions in Guptapara (47 particles, 45.2%) but reduced prevalence in Dignabad (19 particles, 12.8%). Films accounted for 268 particles (20.89%), with Janglighat recording the highest concentration (100 particles, 27.0%) and Guptapara exhibiting none. Foam particles (195 total, 15.20%) were most abundant in Chatham (74 particles, 19.2%) and absent in Chidiyatapu, while paint particles (183 total, 14.26%) demonstrated higher relative contributions in Dignabad (35 particles, 23.5%). These findings underscore the dominance of fragments and fibers regionally, alongside stark site-specific disparities, such as Guptapara's complete lack of films and Chidiyatapu's foam scarcity, suggesting differential anthropogenic inputs or degradation pathways. A Mann–Whitney U test comparing microplastic shape distributions between the North (Chatham, Jungli Gut, Dignabad) and South (Wandoor, Guptapahar, Chidiyatapu) regions revealed statistically significant differences for multiple morphotypes. Foam ($p < 0.001$), paint particles ($p < 0.001$), and film particles ($p = 0.0048$) were all significantly more abundant in the Northern region. Specifically, foam and paint particles contributed 20.55% and 16.69% of the total microplastics in the North, respectively, but only 2.38% and 8.47% in the

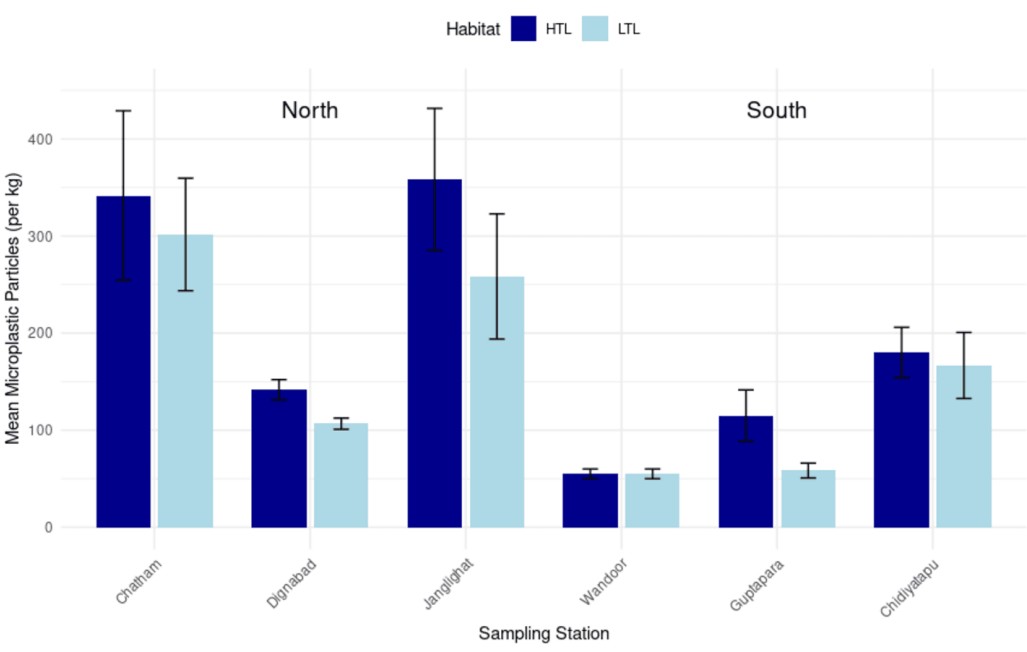

**Figure 3  Comparison of microplastic concentrations (particles per kg) in sediment across sampling stations, highlighting differences between high tide line (HTL) and low tide line (LTL) habitats.** The mean concentration of microplastic particles (per kg) observed at different sampling stations located in the North and South regions, categorized by habitat type: HTL (high tide line) and LTL (low tide line). Dark blue bars represent HTL values, while light blue bars indicate LTL values. Error bars denote standard deviations, reflecting variability within samples. Among northern stations, Janglighat and Chatham showed the highest microplastic concentrations, especially in the HTL zones, while Dignabad and Wandoor had relatively lower levels. In the south, Chidiyatapu exhibited the highest mean concentration, followed by Guptapara and Wandoor. Generally, HTL zones recorded higher mean microplastic particle counts than LTL zones across most stations, highlighting the influence of tidal deposition patterns on microplastic accumulation.

South. This pattern suggests spatial heterogeneity in microplastic inputs or retention dynamics, likely linked to localized anthropogenic sources such as urban runoff, paint degradation, or packaging waste. In contrast, fragments ($p = 0.078$) and fibers ($p = 0.287$) did not differ significantly between regions, although their higher proportions in the South (39.42% and 33.07%, respectively) may reflect ongoing inputs from textile washing, fish processing, or plastic degradation. The complete absence of film particles in Guptapahar and the near-total scarcity of foam in Chidiyatapu further underscore localized variation in microplastic composition, driven by both environmental and anthropogenic factors. However, contrary to initial expectations, the Mann–Whitney U test results showed no statistically significant differences ($p > 0.3$ for all shape types) in microplastic shape abundance between HTL and LTL zones. This suggests that tidal level (HTL *vs.* LTL) may not have a strong influence on the shape-based distribution of microplastics. While some trends such as a relative increase in fragments and fibers in LTL habitats, and higher films and foams in HTL zones were visually observable, these differences were not statistically significant, possibly due to small sample size or high intra-zone variability. Therefore,

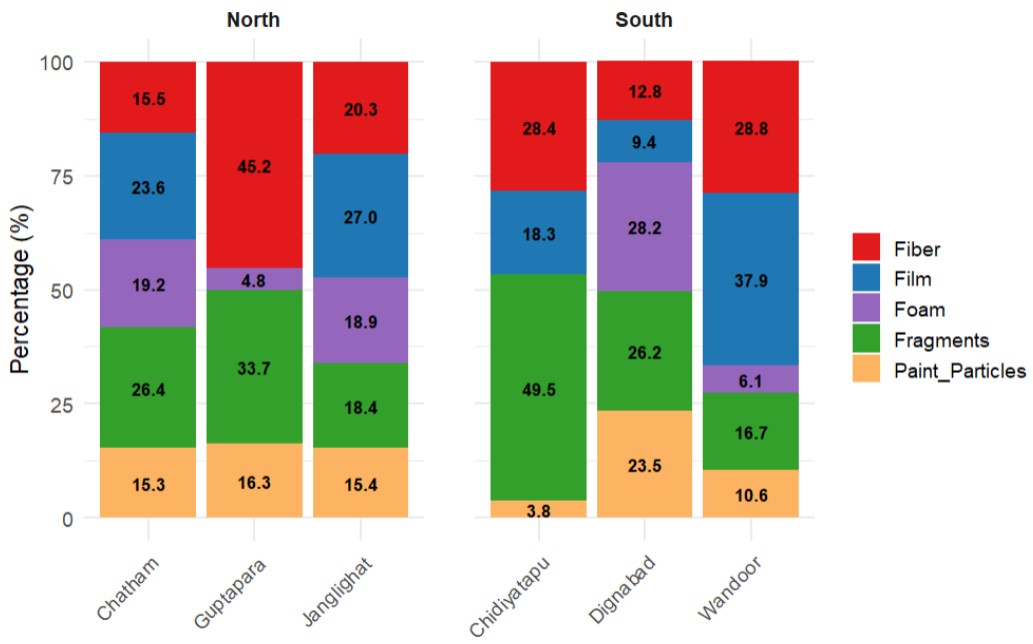

**Figure 4** Percentage composition of microplastic particle types across six coastal locations in South Andaman. The relative proportions of fiber, film, foam, fragments, and paint particles at each location. Fragments dominate at Chidiyatapu (49.5%) and Guptapara (33.7%), while Film is most abundant at Junglighat (27%) and Wandoor (37.9%). Fiber contributes significantly at Guptapara (45.2%), whereas paint particles and foam remain relatively low across all sites. This distribution highlights distinct microplastic profiles between northern (left group) and southern (right group) locations.

**Table 1** Abundance and shape distribution of microplastics (particles per 200 g) across study sites.

| S. No. | Site | Total Number of MP | | Shape | | | | |
|---|---|---|---|---|---|---|---|---|
| | | HTL | LTL | Film | Fragments | Fiber | Foam | Paint particles |
| 1. | Chatham | 205 | 181 | 91 | 102 | 60 | 74 | 59 |
| 2. | Dignabad | 85 | 64 | 14 | 39 | 19 | 42 | 35 |
| 3. | Janglighat | 215 | 155 | 100 | 68 | 75 | 70 | 57 |
| 4. | Wandoor | 33 | 33 | 25 | 11 | 19 | 4 | 7 |
| 5. | Guptapara | 69 | 35 | 0 | 35 | 47 | 5 | 17 |
| 6. | Chidiyatapu | 108 | 100 | 38 | 103 | 59 | 0 | 0 |

**Notes.**
MP, Microplastics; HTL, High tide line; LTL, Low tide line.

microplastic shape distribution appears to be more influenced by site-specific activities and sources than by tidal zone alone.

Colour analysis revealed white (14.03%) and red (13.80%) as the most pervasive hues, collectively constituting over a quarter of all particles (Fig. 5). White dominated in Wandoor (30.3%) and Guptapara (23.1%) in Southern region, while red prevailed in Janglighat (15.4%) in northern region and Chidiyatapu (16.8%) in southern region. Secondary contributors included ash (11.93%), purple (11.85%), and green (11.22%),

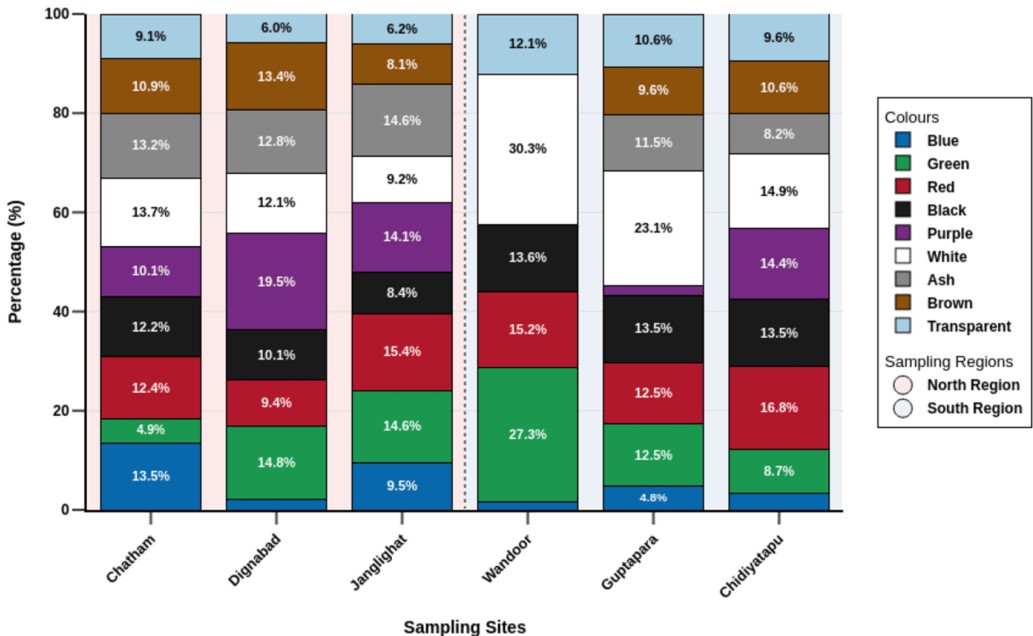

**Figure 5  Comparison of microplastic color composition across sampling sites in the North and South regions.** The percentage composition of various coloured items (blue, green, red, black, purple, white, ash, brown, and transparent) collected from six sampling sites across two regions of the Andaman Islands: North Region (Chatham, Dignabad, Junglighat) and South Region (Wandoor, Guptapara, Chidiyatapu). Each stacked bar represents a site and shows the relative abundance of each colour category. Transparent items dominate in Wandoor (30.3%), while White is most prevalent in Guptapara (23.1%). The variation in colour distribution across sites provides insights into regional differences in debris composition.

which demonstrated marked spatial variation. Ash peaking in Janglighat (14.6%), purple in Dignabad (19.5%) both in northern region, and green in Wandoor (South) (27.3%). Black (11.22%) and brown (9.66%) showed moderate contributions, though black was elevated in Guptapara and Chidiyatapu (13.5% each), while brown remained scarce except in Dignabad (13.4%). Blue (8.03%) and transparent particles (8.26%) were globally minor but exhibited localized relevance, with blue prominent in Chatham (13.5%) and transparent particles marginally higher in Wandoor (12.1%) and Guptapara (10.6%). Geographical scarcity patterns emerged distinctly, purple was absent in Wandoor, blue fell below 4% in Dignabad, Guptapara, and Wandoor, while ash and brown were undetectable in Wandoor. The above results indicate a spatial heterogeneity of colour dominance with white and red as ubiquitous players, green /purple local players and blue/transparent are a function of site related anomalies. This variability indicates that the spatial distribution of microplastic colors may be influenced by site-specific anthropogenic activities (use of colored packaging, fishing gear and fishing activities), hydrodynamic conditions such as tidal forces, and sediment retention characteristics, which together affect the transport, sorting, and deposition of colored particles. To statistically assess regional variation in microplastic colour distribution, Mann–Whitney U tests were conducted for each colour between Northern and Southern regions. The analysis revealed significant differences for

select colours, including blue, purple, and brown ($p < 0.05$), suggesting that these hues vary meaningfully between regions. In contrast, other colours such as red, white, and transparent did not exhibit statistically significant differences ($p > 0.05$), indicating a more uniform presence across both regions. These findings imply that regional factors such as local anthropogenic pressures, tidal regimes, and sediment retention characteristics may influence the deposition patterns of specific microplastic colours. While red and white appeared more evenly distributed, colours like green, blue, and purple likely reflect localized sources or distinct regional transport mechanisms. In addition to regional disparities, habitat-specific analyses across HTL and LTL zones were conducted to examine differences in microplastic colour composition. HTL zones appeared to contain a higher proportion of visually conspicuous microplastics, particularly red (16.65%), green (14.06%), and purple (13.55%), while LTL zones were marked by greater relative contributions of ash (14.00%), brown (13.33%), and transparent particles (10.67%) compared to their respective HTL proportions of 9.42%, 8.00%, and 5.55%. However, Mann–Whitney U tests revealed no statistically significant differences in individual colour distributions between HTL and LTL zones ($p > 0.05$ for all colours), suggesting that while compositional patterns vary visually, these differences are not strong enough to be considered statistically distinct at the scale of this study. These results imply that factors such as strandline accumulation and hydrodynamic sorting may influence microplastic deposition, but their effects on colour-specific distribution across tidal zones may be more subtle or variable than initially assumed.

## Polymer identification

A total of 34 polymer particles were identified across all study locations, categorized into various types. Although only a sub-sample of 34 particles was analyzed for polymer composition, it was deliberately selected to represent diverse microplastic categories across shape, color, location, and tidal zones. This allowed for a preliminary assessment of polymer diversity and potential anthropogenic sources despite logistical limitations. The most abundant polymer was Aramid Fiber (38.24%) and found at multiple locations except Chatham and Guptapara. Aramid fibers, commonly used in industrial and domestic applications like protective clothing and filtration systems, highlight significant anthropogenic contributions. The second most prevalent polymer was acrylonitrile-butadiene rubber (11.76%) identified in Chidiyatapu and Dignabad. This polymer, widely used in automotive and industrial products, suggests a strong link to transport or industrial waste runoff. Similarly, polyisoprene (11.76%), was detected in Wandoor and Dignabad. Its presence is indicative of discarded rubber products such as gloves, footwear, or tires. Acrylonitrile (8.82%), found in across Chatham, Wandoor, and Chidiyatapu, reflects its localized sources, possibly from synthetic fibers or plastics used in consumer or industrial goods. While other polymers such as acrylic fiber, polyethylene glycol, and cellulose were present in smaller quantities, their lower abundance points to more specific or localized usage patterns. These findings underscore the dominance of industrial and consumer waste in contributing to microplastic pollution, particularly in regions with higher anthropogenic activity.

## Drivers of microplastic distribution

Spatial analysis revealed distinct differences in microplastic shape composition, land cover characteristics, and anthropogenic influence across the fish landing centers. Non-metric multidimensional scaling (NMDS) analysis demonstrated clear clustering based on microplastic shape types, with a notable separation between northern and southern regions (Fig. 6), while samples from HTL and LTL zones showed partial overlap, indicating minimal distinction. The NMDS model yielded a stress value of 0.1396, indicating a reliable two-dimensional ordination. PERMANOVA results supported the regional separation, revealing a statistically significant difference in shape composition between northern and southern regions (pseudo-F = 16.013, $R^2$ = 32.02%, $p = 0.001$). In contrast, the difference between HTL and LTL zones was not statistically significant (pseudo-F = 0.7265, $R^2$ = 2.09%, $p = 0.556$), suggesting that tidal position had a limited influence on microplastic shape distribution. These findings indicate that the types of microplastic particles differ notably between regions and are also influenced by tidal zones, with HTL and LTL zones exhibiting distinct clustering patterns. LULC analysis (Fig. 7) revealed considerable variation in built-up area across sampling locations. Northern fish landing centres had substantially higher built-up area within 1 km buffer zones: Dignabad (58.73%), Janglighat (47.03%), and Chatham (44.51%). In contrast, southern stations such as Wandoor (0.24%), Chidiyatapu (0.10%), and Guptapara (0.05%) exhibited minimal built-up coverage. Supplementary Material 3 provides the complete LULC classification used for this assessment. PCA was conducted to assess the relationships among microplastic abundance, population density, and land cover (Fig. 8). PC1 and PC2 accounted for 63.6% and 18.8% of the variance, respectively, cumulatively explaining 82.4% of the total variability. The PCA biplot showed that microplastic abundance was positively associated with built-up area and population density, as indicated by the direction and alignment of the loading vectors. Northern sites clustered closer to these anthropogenic variables in the PCA space, while southern sites were situated in areas dominated by forest cover, mangroves, or open water features. Supplementary Material 4 presents the full PCA variable loadings and site coordinates.

## DISCUSSION

### Spatial distribution and anthropogenic influences on microplastic abundance

Microplastics were ubiquitously detected across all sampling stations, with concentrations varying between 10 and 85 particles per 200 g of sediment, equating to 50–425 particles per kg. The average concentration stood at 178.15 ± 112.40 particles per kg, demonstrating significant spatial variability influenced by localized anthropogenic activities. The highest concentration, recorded at Chatham station (321.65 ± 69.85 particles/kg), underscores the influence of localized anthropogenic activities such as ship docking, dense urbanization, and high commuter traffic. Chatham, with a population of approximately 13,922 in an area of 3.14 km$^2$ (population density ~4,430 people/km$^2$), accounts for nearly 12.9% of the total Port Blair Municipal Council population (2011 Census of India; *Directorate of Census Operations, Andaman & Nicobar Islands, 2014*). Additionally, the Chatham ferry

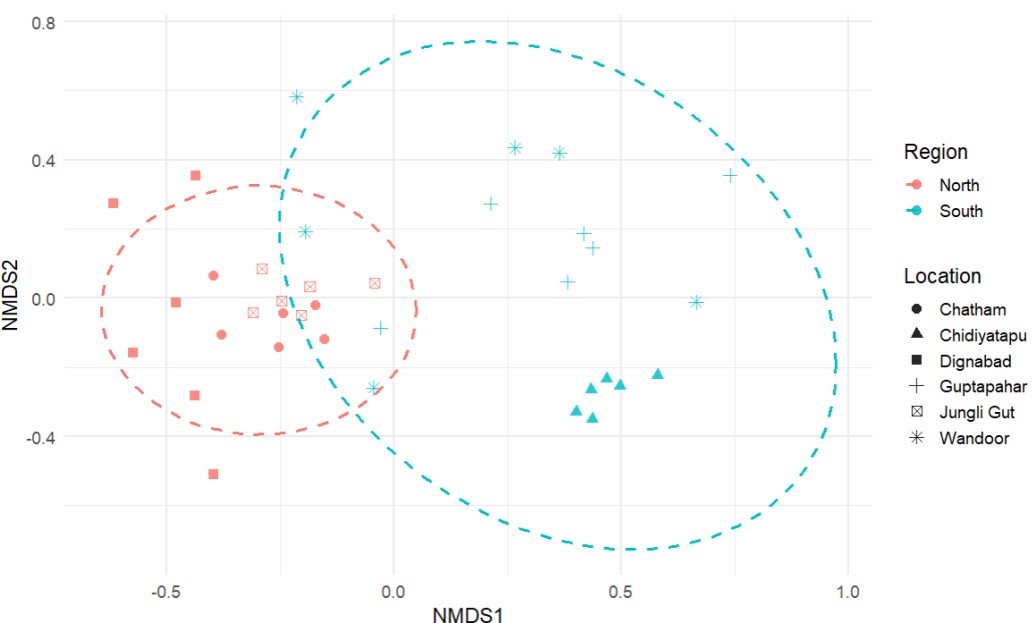

**Figure 6  NMDS ordination based on Bray–Curtis similarity matrices from microplastic shape composition data from the northern and southern regions of the study area.** Each point represents a sample, categorized by location using distinct shapes and by region (North in red, South in blue) using color. The dashed ellipses represent 95% confidence intervals around samples from each region, indicating the degree of similarity within groups. The clear separation between the North and South ellipses suggests significant differences in microplastic composition between these regions. Northern sites (Junglighat, Dignabad, Chatham) show tighter clustering, whereas southern sites (Wandoor, Guptapahar, Chidiyatapu) are more dispersed, indicating greater variability in microplastic characteristics.

jetty serves as a major transit hub between islands for tourism and industrial transportation, accommodating an estimated 20,000 daily commuters. The detection of polymers such as acrylonitrile–butadiene rubber, aramid fibers, acryl fibers, polyethylene glycol, and polyisoprene suggests inputs from industrial materials, urban textiles, transportation infrastructure, and domestic waste. These diverse polymer types point to mixed land-based and maritime-associated sources contributing to heightened microplastic accumulation in the surrounding sediment. Shipping operations, characterized by the release of paint particles, synthetic ropes, and plastic waste, alongside urban runoff from populated areas, likely contributed to this elevated abundance (*Dowarah & Devipriya, 2019*), also identified fishing and maritime activities as key drivers of microplastic proliferation. In contrast, Wandoor station exhibited the lowest concentration (56.00 ± 4.45 particles per kg), attributed to its status as a national park with stringent waste management protocols, including regulated plastic disposal and efficient waste collection systems. This finding reflects a notable reduction from the 345.0 ± 57.8 particles per kg (*Patchaiyappan et al., 2020*) in Wandoor's beach sediments, suggesting progressive improvements in waste management and community awareness over time, as corroborated by *Mohan et al. (2022)*, who documented declining microplastic levels between 2019 and 2023. Regional comparisons further emphasize this variability: microplastic abundances in Tamil Nadu

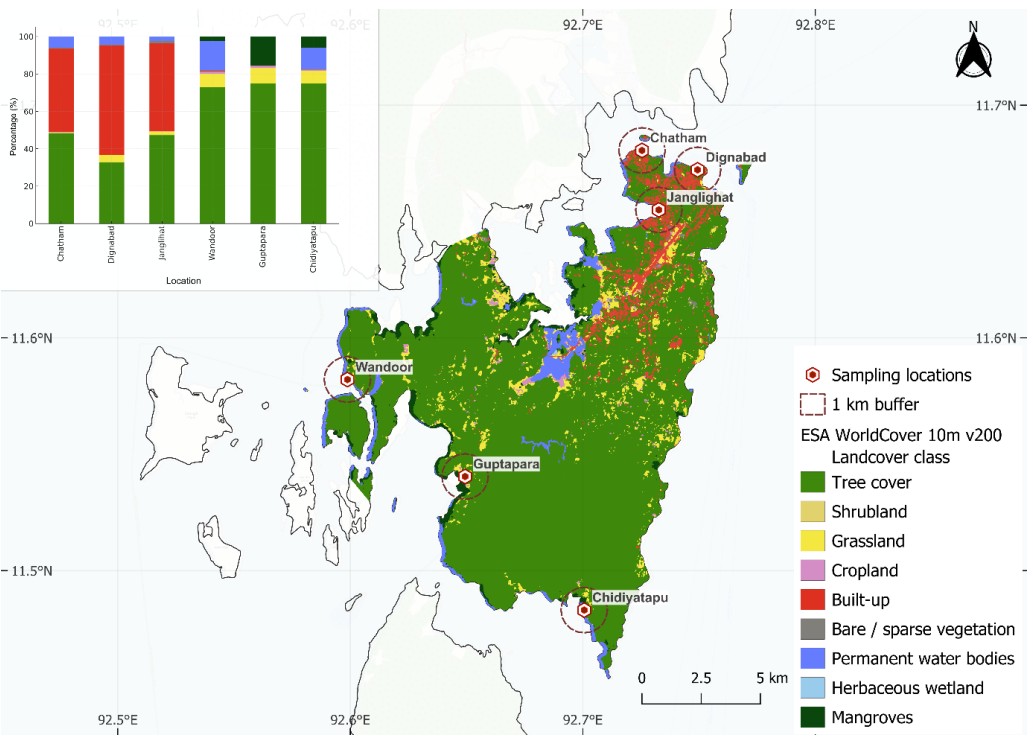

**Figure 7  Land cover composition around fish landing sites in South Andaman (ESA WorldCover 10 m).** Land use and land cover (LULC) around fish landing centers in South Andaman using ESA World-Cover 10 m v200 data. Each red hexagon marks a sampling site, encircled by a 1 km buffer zone. Colors represent different LULC classes such as tree cover, cropland, built-up areas, wetlands, and water bodies. The inset bar plot shows the percentage of each land cover type within the buffer zones, highlighting spatial differences between urbanized and vegetated areas.

(181 ± 60 particles per kg) (*Tiwari et al., 2019*) parallel the current study's average, whereas South Andaman sediments exhibited lower concentrations (45.17 ± 25.23 particles per kg), likely due to disparities in human activity intensity, regulatory frameworks, and environmental policies (*Goswami, Vinithkumar & Dharani, 2020*). Such spatial heterogeneity highlights the necessity for context-specific mitigation strategies to address microplastic pollution. A comparison with other similar studies is presented in Table 2. Microplastic pollution is prevalent across both urban and rural zones. Urban fish landing centres such as Junglighat, Chatham, and Dignabad serve as hotspots due to high plastic use and inadequate waste management systems (*Shankar et al., 2024*; *Patchaiyappan et al., 2020*). Improperly disposed plastics from these centres, combined with urban runoff and industrial activities such as boat maintenance, contribute to marine debris (*Jaini & Namboothri, 2022*). Meanwhile, rural coastal zones like Guptapara, Wandoor, and Chidiyatapu, which are closer to coral reefs and mangroves, also exhibit microplastic contamination. Urban runoff, domestic waste, and transboundary debris transported by ocean currents appear to be major contributors in these regions, as suggested by the presence of diverse polymers such as synthetic rubbers, aramid and acryl fibers, and polyethylene derivatives (*Robin et al., 2019*; *Rochman et al., 2015*).

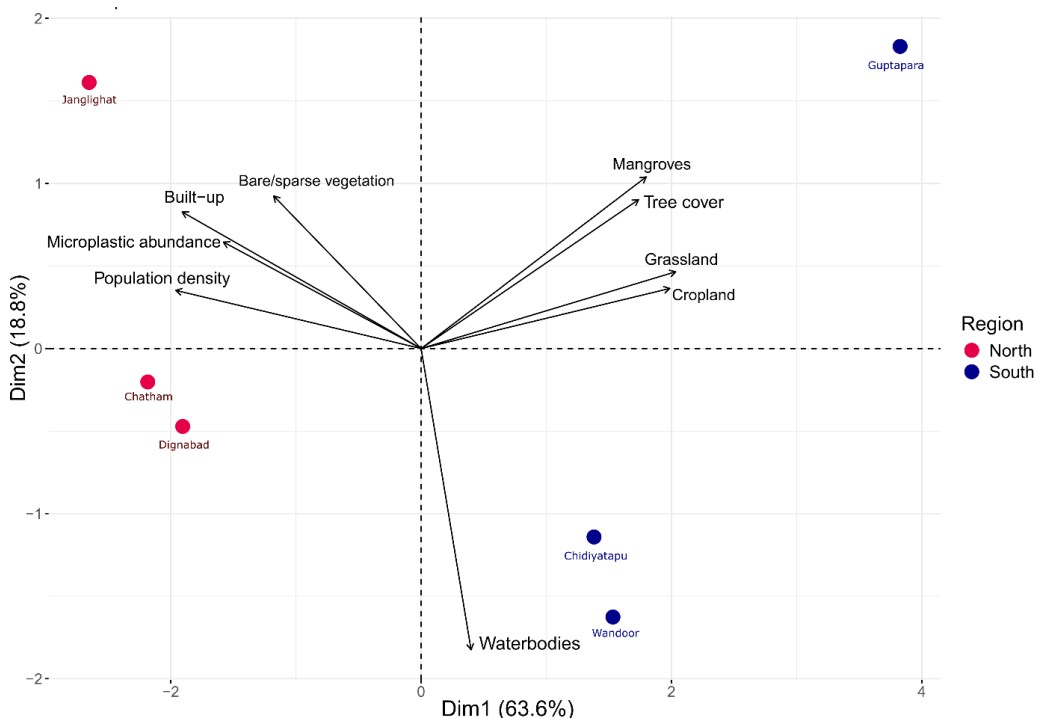

**Figure 8** Principal component analysis (PCA) biplot illustrating the relationships between microplastic abundance, land use and land cover (LULC) composition, and population density. The spatial distribution and grouping of sampling sites based on two principal components (PC1 and PC2). Red dots represent urban-influenced sites (*e.g.*, Junglighat, Chatham, Dignabad), while blue dots represent rural or relatively pristine sites (*e.g.*, Chidiyatapu, Wandoor, Guptapara). The separation along PC1 and PC2 suggests variation in environmental parameters or biological communities between these groups. The arrows indicate the loading vectors of the original variables, showing how they contribute to each principal component. Sites clustering together share similar characteristics, and the direction and length of the arrows reflect the influence of variables on site grouping.

## Morphological and chromatic characteristics of microplastics: sources and ecological risks

Microplastic particles were classified into five morphological shapes based on the films (20.89%), fragments (27.9%), fibers (21.75%), foam (15.20%), and paint particles (14.26%) forms of plastic debris. The distribution was main dominated by fragments the result of continual fragmentation of large plastic debris by abrasion, UV radiation and biological weathering. High tourist footfall in South Andaman's beaches exacerbates plastic waste generation, particularly from single-use items like cups, straws, and packaging materials, leads to pollution, as Andaman has inadequate recycling infrastructure and improper waste disposal practices (*Central Pollution Control Board (CPCB), 2016*). These plastics are transported to coastal areas *via* oceanic currents, wind, and riverine systems, where prolonged environmental exposure fragments them into microplastics (*Carson et al., 2012*; *Kim et al., 2015*; *Karthik et al., 2018*). The persistence of fragments in sediments raises ecological concerns, as their durability enables long-term retention, leaching of toxic additives (phthalates, bisphenol A), and adsorption of hydrophobic pollutants (*Cole*

**Table 2  The similar studies on microplastics assessment in sediment in India.**

| Location | Sample type | Shape | Colour | Polymer type | Polymer identification | Abundance | Refrences |
|---|---|---|---|---|---|---|---|
| South Andaman | Sediment | Fragments, Fibre, Film, Foam, Paint Particle | White, Transparent, Blue, Brown, Black, Purple | Polyisoprene, NBR, Polybutadiene, PEG, PVDC, Pinene, Cellulose | ATR-FTIR | 178.2 ± 103.3 particles/kg | This study |
| South Andaman Islands | Sediment | Fragment, fibre, spherule | white, transparent, grey, red/orange, yellow, green, blue, violet black | PP, PVC | Raman spectroscopy | 414.35 items/kg | *Patchaiyappan et al. (2020)* |
| Andaman Sea continental shelf | Seafloor sediment | fiber, fragment, and pellet | blue, transparent, red, black, yellow, green, brown, and pink | Acryl fiber, PE, PEP, and Nylon | ATR-FTIR | 0–38 items/kg | *Goswami, Vinithkumar & Dharani (2021)* |
| Arabian Sea continental shelf | Seafloor sediment | fiber, fragment, and pellet | Blue, Transparent, Red, Black, yellow, green, brown, and pink | Acryl fiber, PE, PEP, and Nylon | ATR-FTIR | 28–267 items/kg | *Goswami, Vinithkumar & Dharani (2021)* |
| Kerala coast | Sediment | Fragments, s, foams, films, and pellets | green, red, yellow, blue and violet | PE, PP, PA, PS, PET, PUR, RY, cellulose | ATR-FTIR | 40.7 ± 33.2 items/ m2 | *Robin et al. (2019)* |
| Maharashtra (Girgaon, Mumbai) | Sediment | Granule, fibre, film | – | PE, PET, PS, PP, PVC, | ATR–FTIR | 220 items/kg | *Tiwari et al. (2019)* |
| Kerala (Nattika Beach) | Sediment | Fragment, fibre, film, Fragment, fibre, film, | – | PE, PP, PS, PE-PP, PCU | ATR–FTIR | 120.85 items/kg | *Ashwini & Varghese (2019)* |
| Tamil Nadu (Tuticorin) | Sediment | Fibre, film, fragment,foam | white, blue, black, green yellow and red | PE, PVC, PP, PS, PET,NY | ATR–FTIR | 25 ± 18 to 83 ± 49 items/m² | *Jeyasanta et al. (2020)* |
| Tamil Nadu Coast | Sediment | Fragments, fibres and foams | – | PP, PE, PS | ATR-FTIR | 1323 ± 1228 mg/ m2 | *Karthik et al. (2018)* |

*et al., 2011*; *Andrady et al., 2022*). Subsequent ingestion by marine organisms facilitates bioaccumulation and trophic transfer, endangering marine biodiversity and human health (*GESAMP, 2015*; *Teuten et al., 2009*). Fibers, the second-most prevalent category, originate predominantly from domestic wastewater containing synthetic textile fibers and discarded fishing gear, reflecting terrestrial and marine anthropogenic contributions (*Browne et al., 2011*; *Robin et al., 2019*). Foam and paint particles, though less abundant, derive from localized sources such as packaging materials and maritime maintenance activities, as observed in Chidiyatapu (*Krishnakumar et al., 2020*). These findings align with global studies documenting fragment predominance in coastal sediments, reinforcing the need

for targeted waste management and policy interventions (*Eriksen et al., 2014*; *Cózar et al., 2014*).

A multi-coloured microplastic palette was detected by color-based analysis: white (14.03%), red (13.8%), ash (11.93%), purple (11.85%), green and black (11.22%), brown (9.66%), transparent (8.26%) and blue (8.03%). Similar trends seen in prior studies are illustrated by the white and red particles recurrence, due to their common aspiration in packaging as well as in the textiles. (*Krishnakumar et al., 2020*; *Syakti et al., 2017*; *Robin et al., 2019*). The microplastics appear brightly colored, making them the perfect mimic of natural prey (planktons) where marine organism has higher likelihood to ingest due their optical similarity. Seabirds and turtles confuse casing and white particles for fish eggs, jellyfish which causes blockages in intestines, chemical exposure as well as starvation (*Wright, Thompson & Galloway, 2013*; *Bugoni, Krause & Petry, 2001*). Fish and crustaceans similarly ingest colored microplastics, mistaking them for prey with potential ramifications for human health *via* seafood consumption (*Chatterjee & Sharma, 2019*). Environmental weathering processes, such as UV exposure, oxidation, and biofouling can significantly alter the original color of microplastic particles by causing fading, yellowing, or surface staining (*Hidalgo-Ruz et al., 2012*; *Abaroa-Pérez et al., 2022*). These color changes reduce the visual detectability of microplastics during manual sorting and can mislead source attribution efforts. Furthermore, altered coloration may affect the way marine organisms interact with microplastics, potentially influencing ingestion rates and ecological outcomes (*Horie et al., 2024*). This highlights the importance of addressing primary sources such as improper waste disposal and synthetic textile release to minimize environmental loading.

Microplastics are a persistent and dominant pollutant in South Andaman's beach and coastal sediments, with average abundances of 414.35 ± 87.4 particles/kg (*Patchaiyappan et al., 2020*; *Mohan et al., 2022*). This accumulation is exacerbated by weak oceanic flushing and constant land-based input, as the Andaman and Nicobar Islands serve as potential catchment areas for ubiquitous marine anthropogenic waste (*Shankar et al., 2023*; *Jambeck et al., 2015*). These particles infiltrate sediments through multiple pathways, including direct beach littering, stormwater runoff, tidal flow, and the fragmentation of larger plastic debris by physical, chemical, and biological processes (*Saravanan et al., 2021*; *Critchell & Lambrechts, 2016*; *Bergmann, Gutow & Klages, 2015*; *Thompson et al., 2004*). Once embedded, microplastics alter sediment texture, porosity, permeability, and oxygen availability, potentially disrupting vital nutrient cycling and microbial processes (*Carson et al., 2011*; *Patchaiyappan et al., 2020*; *Zettler, Mincer & Amaral-Zettler, 2013*). Furthermore, microplastic particles in sediment act as carriers, binding hazardous compounds such as heavy metals, polychlorinated biphenyls (PCBs), polyaromatic hydrocarbons (PAHs), and phthalates, increasing the risk of contaminant mobilization during sediment disturbance (*Goswami, Vinithkumar & Dharani, 2020*; *Allen et al., 2018*; *Tang et al., 2018*). This contamination poses a severe threat to benthic communities, as infaunal species like worms, mollusks, and corals can ingest these particles, leading to physical injury, inflammation, reduced feeding, compromised growth, and reproductive output (*Gall & Thompson, 2015*; *Van Cauwenberghe et al., 2015*; *Wright, Thompson & Galloway, 2013*; *Sussarellu et al., 2016*; *Hall et al., 2015*). Despite increasing awareness, soil
sediments at fish landing centers are an often-overlooked reservoir of microplastics, and comprehensive, quantified data on marine debris in remote Indian islands has historically been scarce (*Kumar et al., 2025*; *Saravanan et al., 2021*; *Goswami, Vinithkumar & Dharani, 2020*; *Krishnakumar et al., 2020*). Therefore, systematic, time-series sediment surveys are critically needed in South Andaman's coastal areas, particularly around fish landing centers, to inform effective coastal conservation strategies, implement targeted waste management practices, and safeguard these fragile benthic ecosystems.

## Anthropogenic and land-use drivers of microplastic distribution

The NMDS analysis, supported by PERMANOVA, revealed clear compositional differences in microplastic shapes between northern and southern fish landing centers, pointing to region-specific pollution sources. This spatial segregation suggests that microplastic input is not homogeneous across the island but instead shaped by localized anthropogenic activities. In the northern centers, the clustering pattern indicated a dominance of fragments, fibers, and paint particles, types often linked to intensive fishing operations, industrial discharges, and urban runoff. These microplastics are likely introduced through boat maintenance, synthetic gear degradation, and poor waste disposal near harbours and residential areas. Conversely, the southern sites showed more constrained and less diverse shape compositions, reflecting reduced plastic input and lower human disturbance. The ordination patterns imply that both the intensity and type of land-based activities such as urban infrastructure, maritime traffic, and tourism are central in influencing the diversity and abundance of microplastics. The PERMANOVA results reinforce that these regional differences are statistically robust, suggesting that future mitigation strategies must consider not only the quantity of microplastic pollution but also its compositional characteristics, which can differ significantly based on surrounding land use and socioeconomic drivers. The polymer types identified in sediment samples from the fish landing centers of South Andaman namely aramid fiber, acrylonitrile–butadiene rubber, polyisoprene, acrylonitrile, polyester, polycarbonate, polypropylene, and polyethylene glycol reveal direct links to localized anthropogenic activities. The high proportion of aramid fiber and polyester, both synthetic textile fibers, suggests inputs from domestic wastewater and discarded fishing gear, as also reported from Surabaya, Indonesia (*Firdaus, Trihadiningrum & Lestari, 2019*) and Car Nicobar Island (*Kiruba-Sankar et al., 2023*). The presence of acrylonitrile–butadiene rubber and polyisoprene, used in tires and fishing gear, aligns with urban runoff and maritime activities (*Patchaiyappan et al., 2020*; *Vaid, Mehra & Gupta, 2021*). Acrylonitrile, associated with industrial discharges, has also been documented in Port Blair Bay sediments (*Goswami, Vinithkumar & Dharani, 2021*). Polypropylene, a common plastic in packaging and fisheries, was similarly found in polluted coastal zones (*Syakti et al., 2017*). The detection of polycarbonate and polyethylene glycol indicates input from fragmented household and industrial plastics (*Robin et al., 2019*; *Falahudin et al., 2019*). These polymer signatures offer strong chemical evidence that microplastic pollution in sites is predominantly driven by anthropogenic sources directly linked to high built-up areas, as revealed by LULC analysis. Together with NMDS and PCA results, they pointing
that urban infrastructure, industrial activity, and maritime operations are the principal contributors to the elevated microplastic load in these regions.

The buffers created around the northern fish landing centres were found to be containing high proportion of built-up area (Fig. 7), which can be ranked in decreasing order of Dignabad (58.73%), Janglighat (47.03%), Chatham (44.51%), Wandoor (0.24%), Chidiyatapu (0.10%), and Guptapara (0.05%) (Supplementary Material 3). The spatial distribution of microplastic abundance across fish landing centres in South Andaman Island demonstrates clear correlations with land use and land cover (LULC) patterns (Fig. 7) and anthropogenic features. Among the northern centres, Chatham shows the highest microplastic loads (321.65 ± 69.85 particles per kg) mainly due to its high built-up area shares (44.51%), and very close connection with a ship docking site. Industrial activities such as ship maintenance and repair contribute microplastics to the marine environment, that is the paint particles and fragments especially this type described inefficient garbage disposal practices implemented in Chatham increase pollution due to industrial and urban waste are often directly discarded to sea, one reason of highly sediment contamination. Dignabad, characterized by a high population density and the most significant built-up area (58.73%) (Fig. 7), experiences substantial household waste mismanagement, with much waste disposed of along the beachside. This unchecked waste disposal is a significant source of microplastic pollution, contributing to the high concentrations observed in sediments.

Similar to this, Junglighat, which is well-known as a fish landing center, is the scene of extensive marine activities, such as boat operations, fish processing, and equipment use, all of which contribute to the microplastic pollution of the nearby sediments. The region's pollution levels are further increased by the essential sources of fibers and pieces found in fishing nets, ropes, and other equipment. Conversely, Wandoor, Chidiyatapu, and Guptapara's southern centers have noticeably lower levels of microplastic, which is consistent with their unique LULC features. These places are mostly covered by natural landscapes like forests and have very little built-up area (Wandoor: 0.24%, Chidiyatapu: 0.10%, Guptapara: 0.05%). Due to strict regulations and well-managed waste disposal systems, Wandoor, which is close to a national park, efficiently reduces pollution. Additionally, Chidiyatapu, a prominent tourist destination, has experienced a significant reduction in pollution levels due to its designation as a forest reserve (Nagesh et al., 2015). This status has facilitated the implementation of enhanced waste management practices enforced by the forest department, contributing to improved environmental sustainability in the region. Because of its sparse urbanization and low population density, Guptapara is subject to less human-induced stresses, which keeps the amount of microplastic in its sediments at a lower level. While locations with a high concentration of natural landscapes, such forests, serve as buffers against contamination, the correlation analysis supports the observed trends, which show a positive association between built-up areas, population density, and microplastic pollution.

Together, PC1 (63.6%) and PC2 (18.8%) explain 82.4% of the total variance (Fig. 8), and microplastic abundance shows a positive correlation with built-up areas and population density (Supplementary Material 4). Fish landing centers that are closer to each other in Euclidean space exhibit similar LULC pattern, microplastic abundance, and population

density. The findings from our study align closely with the conclusions of previous research, which demonstrated that population density is positively correlated with microplastic abundance (*Kunz et al., 2023*; *Stolte et al., 2015*). Our results indicate that highly populated areas such as Chatham, Dignabad, and Junglighat exhibit the highest microplastic concentrations. This can be attributed to dense populations and intense industrial and commercial activities, as revealed by the LULC (Fig. 7) analysis. These findings align with previous studies reporting elevated microplastic levels in densely populated coastal regions, emphasizing the influence of urbanization and population growth particularly given that nearly 80% of marine plastic waste originates from land-based sources. Another study reported elevated microplastic concentrations in coastal regions with high population densities, supporting our findings that northern locations with concentrated human activities experience higher levels of microplastic pollution (*Bobchev et al., 2024*). Furthermore, the *European Commission (2011)* reported that approximately 80% of marine plastic waste originates from land-based sources, emphasizing the role of urbanization and population growth in contributing to microplastic pollution.

In northern regions like Junglighat, which serves as a hub for major marine activities, population density, industrialization, and improper waste management were found to be critical factors influencing microplastic abundance. Similar observations were made by *Chen et al. (2023)* and *Vaughan, Turner & Rose (2017)*, who noted that growing populations increase the demand for plastic products, with inadequate waste management practices serving as a major source of microplastic pollution in coastal areas. The PCA results (Fig. 8) revealed a strong correlation between microplastic abundance, population density, and built-up areas at sites such as Chatham, Dignabad, and Junglighat. The arrows representing microplastic abundance, built-up areas, and population density in the PCA plot point in the same direction, highlighting their interdependence. Northern locations are positioned closer to these variables, reflecting their higher values, while southern regions, characterized by mangroves, water bodies, and tree cover, showed comparatively lower microplastic levels.

## CONCLUSIONS

This study reveals significant spatial heterogeneity in microplastic (MP) pollution within soil sediments of fish landing centers across Sri Vijaya Puram (Port Blair), driven primarily by localized anthropogenic activities. Northern regions, exemplified by hotspots like Chatham and Junglighat, exhibited elevated MP concentrations linked to dense urbanization, intensive maritime traffic (*e.g.*, ship docking, fishing), and inadequate waste management. These areas showed strong correlations between MP abundance and urban runoff, industrial discharges, and degraded fishing gear. In contrast, southern sites such as Wandoor, situated within natural landscapes with lower anthropogenic pressure, demonstrated reduced contamination, attributed to effective waste management practices and limited industrial activity. Multivariate analyses, including NMDS and PERMANOVA, confirmed pronounced regional differences in MP shape composition, underscoring the influence of divergent land-use patterns. Polymer fingerprints further implicated

industrial and consumer waste inputs, with dominant polymer types (*e.g.*, polyethylene, polypropylene) and fragmented shapes aligning with degraded packaging, textiles, and maritime equipment. Principal component analysis reinforced that built-up areas and population density strongly correlate with MP abundance, highlighting urbanization as a critical driver. The stark north-south divide emphasizes the disproportionate impact of human settlement and economic activities on coastal MP pollution. High-risk northern zones require prioritized interventions, including upgraded waste infrastructure, stricter maritime regulations, and urban runoff mitigation. Conversely, southern regions illustrate the efficacy of conservation-oriented practices. These findings underscore the necessity for spatially targeted policies addressing sector-specific sources (fishing, shipping, urban waste) and adaptive management frameworks to reduce plastic leakage in tropical coastal ecosystems. The study provides a template for mitigating MP pollution in analogous settings, advocating for integrated strategies that reconcile ecological preservation with anthropogenic development. Future studies should prioritize long-term temporal monitoring to assess seasonal or annual MP trends, coupled with advanced source apportionment techniques (isotopic tracing) to identify precise pollution pathways. Vertical sediment profiling could elucidate MP sequestration dynamics, while ecotoxicological assessments are essential to evaluate risks to endemic biota. Additionally, comparative evaluations of waste management efficacy across regions could inform context-specific mitigation strategies. Addressing these gaps will enhance predictive models and support evidence-based policymaking to preserve vulnerable island ecosystems from escalating plastic pollution.

## ACKNOWLEDGEMENTS

This study was conducted as internship work of Miss Akshatha Soratur at ICAR-CIARI. The authors sincerely thank the Director of ICAR–CIARI and the Officer-in-Charge of ACOSTI-NIOT, Sri Vijaya Puram (Port Blair), for their support and encouragement in completing this research. We also extend our gratitude to the staff of the Division of Fisheries Science, ICAR-CIARI, for their help with sample collection.

### Funding

The authors received no funding for this work.

### Competing Interests

Balu Alagar Venmathi Maran is an Academic Editor for PeerJ.

### Author Contributions

- Ajit Kumar performed the experiments, analyzed the data, prepared figures and/or tables, authored or reviewed drafts of the article, and approved the final draft.
- Akshatha Soratur performed the experiments, analyzed the data, prepared figures and/or tables, authored or reviewed drafts of the article, and approved the final draft.

- Sumit Kumar analyzed the data, prepared figures and/or tables, authored or reviewed drafts of the article, and approved the final draft.
- R Kiruba-Sankar conceived and designed the experiments, authored or reviewed drafts of the article, and approved the final draft.
- Dilip Kumar Jha analyzed the data, authored or reviewed drafts of the article, and approved the final draft.
- Balu Alagar Venmathi Maran conceived and designed the experiments, authored or reviewed drafts of the article, and approved the final draft.

## Data Availability

The data is available in the Supplementary Files.

## Supplemental Information

Supplemental information for this article can be found online at http://dx.doi.org/10.7717/peerj.19965#supplemental-information.

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
