# Peer review of "Drivers of microplastic pollution in soil sediments at fish landing centers in Sri Vijaya Puram (Port Blair), South Andaman Island"

_PeerJ, doi:10.7717/peerj.19965_

## Round 0.1 · original submission · Major Revisions

· Academic Editor

Major Revisions

You are requested to diligently address all the comments. In case, you don't agree with a comment of a reviewer, justify that with reasoning.

**Language Note:** The review process has identified that the English language must be improved. PeerJ can provide language editing services - please contact us at [email protected] for pricing (be sure to provide your manuscript number and title). Alternatively, you should make your own arrangements to improve the language quality and provide details in your response letter. – PeerJ Staff

Reviewer 1 ·

Basic reporting

The language employed by the author is suitable for scientific article writing.
The tables and figures included in the manuscript are comprehensive and properly cited.
The manuscript adheres to the established systematics for writing scientific articles, in accordance with the author guidelines provided by the journal.
Furthermore, the raw data is accessible and meticulously recorded. Regarding the PCA, is the LULC data for the fishing landing center classified as built-up? A clarification on this point would be appreciated.

Experimental design

The research and articles presented align with the aims and scope of the journal, particularly within the field of Environmental Science. The research question is clearly articulated; however, it would benefit from a more comprehensive identification of the knowledge gap.
The investigation is thorough and detailed, encompassing the extraction of samples, observation, validation of microplastics, and data analysis, all of which are adequately conducted to meet the research objectives.
Nonetheless, further clarification is needed regarding the mention of "parametric and non-parametric tests." An explanation of this statement would be appreciated.

Validity of the findings

What is the importance of examining the drivers or potential sources of microplastics? This approach should be used to differentiate findings from previous studies. In the context of HTL and LTL groupings, this concept introduces a novel perspective, highlighting environmental factors that influence microplastic distribution.
It is essential to investigate whether the variations in shapes and colors of microplastics differ between Northern and Southern regions. Additionally, it is important to determine if these variations align with comparative analyses among different regions and habitats. Habitat-specific analyses have indicated that microplastics tend to be more prevalent in HTL environments compared to LTL environments. Thus, it is pertinent to explore how the shape and color of microplastics may vary both regionally and across different habitats.
Is it possible to modify the NMDS analysis result graph? Please kindly visit this link: https://www.sciencedirect.com/science/article/pii/S0043135422000793
https://www.researchgate.net/publication/353004009_Microplastic_contamination_in_Great_Lakes_fish
Figures 6, 7, and 8 should be incorporated into the Results section, considering the distinct separation of Results and Discussion in this manuscript. Please kindly add more detailed discussion pertaining to these results.
What urban activities significantly contribute to the input of microplastic in sediment?

Additional comments

Please take note of the following technical issues:
1. On lines 133-134, the font color of the citation "Dowarah and Devipriya (2019)" should be formatted according to guideline.
2. Lines 198-206 duplicate the content found in lines 157-165; please amend this to eliminate redundancy.
3. On line 318, the font color of text "Table 2", please adjust according to guideline

Annotated reviews are not available for download in order to protect the identity of reviewers who chose to remain anonymous.

·

Basic reporting

The manuscript entitled “Drivers of microplastic pollution in soil sediments at
fish landing centers in Sri Vijaya Puram (Port Blair), South Andaman Island” focuses on of microplastics abundance and characteristic along with pollution load in sediment at fish landing points in Sri Vijaya Puram (Port Blair), South Andaman Island.

The topic is fair. However, based on scientific consideration, the manuscript contains some findings that may contribute knowledge in the field. Generally, the manuscript quality is acceptable, however, it contains some unclear points that the authors must addressed. The existing literature and discussion are not properly cited and out date. The results and discussion part need to be improved. Below are some suggestions, which hopefully can help the authors improve this manuscript.

I suggest a recommendation for clarifying and tightening up the writing style, but the entire manuscript would benefit from a careful editing. This manuscript has been accepted as major revision.

Experimental design

ok

Validity of the findings

The polymer types provide valuable information but do not clearly show the connection to anthropogenic activities. So, the author should find further lab results to strengthen the manuscript.
Figure2: What is unit of microplastic abundance? And please check the results (The North = 251.4 ± 110.3 particles/kg from the context)
Figure4: The unit should represent in percentage
1. Line259-274: all color name should be in small capital
2. Line273-274: Please clarify this sentence. How does color of microplastic affected by the prescribed factors?
3. Line277: The number of sub-sample 34 polymer particles is not enough (total number of plastics > 1000 pieces)
4. Table1: Please check number of microplastics at Chidiyatapu and add unit.
5. Line300-301: Please represent the number of populations around Chatham station to confirm the conclusion.
6. Line302-305, 325-326: How to approve these sentences because the polymer types from the previous section different from microplastics from fishing activities.
7. Line342: Please check that “Barboza et al., 2018” studied about microplastics and food safety.
8. Line363-367: Please rewritten this part, how relate with color?
9. Line368-385: This part is quit out of text. The author should discuss in the content of microplastic in sediment.
10. Line439-448, 455-457: Please consider rewriting this part to be more concise.

Additional comments

-

---

## Round 0.2 · accepted · Accept

· Academic Editor

Accept

I congratulate you on acceptance of your manuscript for publication by the reviewers. However, the figures need to be made more prominent by enlarging figure axes and titles.

Reviewer 1 ·

Basic reporting

Good

Experimental design

Well done

Validity of the findings

Good

·

Basic reporting

I thank the authors for carefully revising the manuscript to address the previous comments and suggestions. The authors had implemented all the required changes requested by the reviewer, and the manuscript is now suitable for publication.

However, there are some point need to be improve about the figures (for examples, axis or label are in small size).

Experimental design

-

Validity of the findings

-

Additional comments

-